# A Comparative Study on the Temporal Effects of 2D and VR Emotional Arousal

**DOI:** 10.3390/s22218491

**Published:** 2022-11-04

**Authors:** Feng Tian, Xuefei Wang, Wanqiu Cheng, Mingxuan Lee, Yuanyuan Jin

**Affiliations:** 1Shanghai Film Academy, Shanghai University, Shanghai 200072, China; 2Shanghai Film Special Effects Engineering Technology Research Center, Shanghai University, Shanghai 200072, China

**Keywords:** EEG, VR, emotional arousal, visual evoked potential

## Abstract

Previous research comparing traditional two-dimensional (2D) and virtual reality with stereoscopic vision (VR-3D) stimulations revealed that VR-3D resulted in higher levels of immersion. However, the effects of these two visual modes on emotional stimulus processing have not been thoroughly investigated, and the underlying neural processing mechanisms remain unclear. Thus, this paper introduced a cognitive psychological experiment that was conducted to investigate how these two visual modes influence emotional processing. To reduce fatigue, participants (*n* = 16) were randomly assigned to watch a series of 2D and VR-3D short emotional videos for two days. During their participation, electroencephalograms (EEG) were recorded simultaneously. The results showed that even in the absence of sound, visual stimuli in the VR environment significantly increased emotional arousal, especially in the frontal region, parietal region, temporal region, and occipital region. On this basis, visual evoked potential (VEP) analysis was performed. VR stimulation compared to 2D led to a larger P1 component amplitude, while VEP analysis based on the time course of the late event-related potential component revealed that, after 1200 ms, the differences across visual modes became stable and significant. Furthermore, the results also confirmed that VEP in the early stages is more sensitive to emotions and presumably there are corresponding emotion regulation mechanisms in the late stages.

## 1. Introduction

Better and more affordable virtual reality (VR) consumer hardware is encouraging a full integration of VR technology with movies, which is progressively leading to a significant advancement in traditional screen cinema [1]. Most people agree that one of the main reasons people utilize media is to satisfy their emotional demands, especially when it comes to entertainment [2]. The processes of emotion and attention are highly connected with both presence and immersion. Several studies have shown that VR can enhance presence, which is directly tied to emotion [3]. However, whether immersion can affect the arousal speed and regulation ability of emotion and attention remains unclear, which highlights the importance of research on the mechanism of brain neural processing in VR.

There is an increasing trend in using immersive VR for more effective simulation. The rationale for this tendency, according to Marin-Morales’ analysis of the literature [4], is that immersive virtual reality (VR) enables researchers to recreate situations under controlled laboratory circumstances with high degrees of presence and interaction and is recognized as an efficient approach for evoking emotion [5]. Several studies have focused on comparing 2D and VR. The majority of investigations utilizing subjective questionnaires concurred that VR can significantly increase user immersion and incorporate more emotional engagement [6]. Kandaurova et al. [7] found that by comparing the subjective evaluation after watching videos in VR and 2D, VR better stimulated empathy and increased responsibility in the viewer, while being able to stimulate viewers to be more willing to contribute money or time to help society. The University of Sheffield [8] allowed subjects to watch a concert video of Mozart’s musical works at high and low immersion levels. A three-dimensional self-report questionnaire was used to assess emotion and the result indicated that participants in the VR environment experienced a significant increase in both pleasure and presence. The acquisition of physiological signals is also increasingly being used in VR research. Chirico et al. [9] compared the emotional arousal of videos for awe in 2D and VR and analyzed galvanic skin response (GSR) and an electromyogram (EMG) to conclude that VR videos can trigger stronger parasympathetic activation, which can effectively enhance the sense of awe and presence in the film. Electrodermal activity (EDA) and heart rate variability (HRV) data were gathered by Higuera-Trujillo et al. [10] to compare 2D, 360, and VR media. Based on the physiological responses of the subjects, their study revealed that VR provided results that were the most similar to reality.

In particular, Hasson et al. in 2008 proposed the concept of Neurocinematography [11], where the effect of a film on its audience can be detected by brain activity. Searching for brain-neural theories to guide artistic development and innovation includes an essential focus on connecting VR cinema to cognitive neuroscience. Recently, more research has focused on neural mechanisms in virtual reality environments, particularly through electroencephalography (EEG) analysis. In a fully non-invasive process with low dangers or restrictions, the EEG is a medical imaging technology that records the electrical activity produced by brain regions on the scalp [12]. The recorded signals can represent the neuro- physiological changes in the brain that occur during cognition and serve as a valid basis for researching information processing, including attention, perception, emotion, movement, decision making, and judgment [13]. To date, there has been limited research using EEGs in VR environments to investigate the effects of emotions. By comparing and contrasting subjective data, EEGs, and skin conductance responses, Tian F et al. [14] examined the impact of VR-2D and VR-3D on emotional arousal (SCR). According to the results, emotional stimuli in a stereoscopic visual environment had a stronger impact on perception and presence and thus aroused emotions to a greater degree. High-frequency brain activity also exhibited brain lateralization. In 2021, they [15] further performed EEGs to explore the effects of different emotional films on viewers in 2D and VR modes. However, these neurological studies of VR all used multimedia products, and there was already research that demonstrated that spatial sound gives users a higher sense of presence in virtual environments [16]. This is more evidence that studies on VR movies should be investigated together and separately to understand how they interact to influence presence and emotions [8]. Therefore, this paper examined the impact of VR vision particularly.

Previous research contrasting VR with conventional 2D has demonstrated that VR films can elicit more robust brain activity and arouse viewers’ emotions more. In the exploration of VR content creation, the rhythm of emotional evocation and attention directing becomes essential to achieve better emotional buildup and rendering to meet the needs of the viewer’s emotional experience. Due to this, it is even more crucial to research the progression of stereo vision’s impact on emotional arousal and attentional allocation. Significant voltage fluctuations caused by evoked brain activity are known as event-related potentials (ERPs). Mental processes including perception, selective attention, language processing, and memory take place across periods of time in the order of tens of milliseconds [12]. ERPs can be used to define the temporal evolution of these activations [17]. P1, P300, and late positive potential are the main sensory visual ERPs used in this kind of emotion research (LPP). As early as the stage of perceptual encoding, this preferential processing of emotional stimuli is apparent [18]: as shown for affective pictures [19] and emotional facial expressions [20], emotional content causes an increase in extrastriate visual cortex activity and modulates the P1 component. The P300 component is defined as a positive peak near 300ms after the stimulus of the event in question [21]. A large number of studies have shown that P300 is associated with many factors such as subjective probability, the importance of the task in question, uncertainty, and attention and emotion [22]. It has been suggested that the time course of the LPP component, which was visible at the occipital to central recording sites, may be a key parameter for indexing emotion regulation [22]. In particular, early ERP components are assumed to reflect early selective visual processing of emotional stimuli, whereas late ERP components are expected to reflect more strategic high-level processes including decision making, response criteria, and prolonged complex analysis of emotional content. More scientific study is required to help guide the rhythm of visual emotion depiction in VR art. However, there is still a lack of research on the time course of emotional arousal in stereoscopic vision.

Stereoscopic visual stimuli are common in VR research, while there is a lack of research on the temporal dimension of impacts on visual processing and emotional arousal. The major purpose of this paper is to assess the effects of traditional 2D and immersion VR vision on the reaction speed and regulatory mechanisms of emotional arousal and attention allocation. Listed below are the relevant research questions (RQs):RQ1: Can stereoscopic visual stimuli affect early visual processing? Does it interact with emotion?RQ2: Is the difference in emotional arousal between 2D and VR-3D stimuli always present and significant?

## 2. Materials and Methods

Before the formal experiment, a pre-experiment was carried out to reduce experimental error. Self-Rating Depression Scale (SDS) and Self-Assessment Manikin (SAM) together with a set of visually induced EEG-based trials made up the formal experiment. We constructed a test system in Unity to conduct EEG experiments in order to investigate the distinct visual language of VR. All the experiments passed the ethical application.

### 2.1. The Participants

In this experiment, 16 healthy volunteers (9 males/7 females) were recruited from Shanghai University. The sample size was selected depending on other studies [20,21] with samples of sixteen persons that showed significant and reliable effects on the modulation of event-related potentials and oscillations. The average age of subjects was 23.25 years (SD = 1.20), and the average years of education were 20.06 ± 0.97 years. Each participant was right-handed, had eyesight that was either normal or corrected to normal, and had no prior history of psychiatric illnesses. A total of 9 out of 16 participants had used virtual reality before, however, none of them had done so during the preceding month. Participants completed two sets of experiments—the 2D group and VR-3D group—randomly in two days to prevent irritation or weariness effects brought on by the lengthy experimental duration. Prior to the experiment, the Self-Rating Anxiety Scale (SAS) and Self-Rating Depression Scale (SDS) were completed. Consent was given voluntarily by all participants.

### 2.2. Experimental Materials and Hardware Equipment

A virtual environment was built using Unity 2020.3.17f1c1 as shown in Figure 1. The scenes were designed to induce positive or negative emotions, and neutral emotions were prepared as a reference for the comparison. The positive environments include a forest in the autumn breeze, a sun-drenched seashore, and a space station in a Chinese New Year atmosphere, while the negative environments include schools of sharks in the deep sea, an impending sandstorm, and a space station on fire. Each emotional video material set contained 24 different environments and was rendered using Unity Recorder, a built-in Unity plugin, for the VR-3D stereoscopic version as well as the 2D desktop version. The video resolution was 4096 × 2048 dpi, the lengths were 6 s, the frame rate was 30 frames per second, and the format was H.264 encoded. The virtual reality environment was created using Unity 2018.4, which was operating on a computer with an NVIDIA GTX 1070D graphics card, 32 GB of RAM, and a 3.4 GHz Intel Xeon E5-1230 V5 processor. An AOC 24-inch LCD served as the computer display. The experimental setting and equipment connections are shown in Figure 2.

### 2.3. Experimental Procedures

For usage in this work, Decety et al.’s [23] experimental paradigm was utilized. The same subject performed the tests split over two days in the VR-3D and 2D groups, respectively, and the order of the two groups was randomized since EEG and VR investigations frequently cause subjects to feel exhausted and may have an impact on the accuracy of the experimental results. As shown in Figure 3, each group consisted of 36 trials, with a random order between each trial, either 24 positive trials and 12 neutral trials or 24 negative trials and 12 neutral trials. Subjects were instructed beforehand to avoid speaking, clenching their teeth, and blinking excessively while viewing the movie in order to limit noise signals. Subjects were instructed to complete the SAM scale for a subjective score after each trial was displayed. The subjects had control over the scoring and the rest intervals in between trials.

### 2.4. Data Recording and Processing

This paper used the Self-Assessment Model (SAM) as a subjectively rated questionnaire that may quantify human emotional reactions to numerous stimuli, including valence and arousal [24]. As seen in Figure 4, the valence score range is 1–9. The greater the number, the more enthusiasm the image inspired in the observer. The range of arousal scores is 1 to 9, with higher scores indicating more arousal.

EEGs were captured using an EEG system from Neuracle (Neuracle Technology, Changzhou, China). EEG recordings were made using common 64-channel Ag/AgCl scalp electrodes that were grounded on the AFz and referenced to the Cpz. The interelectrode impedances were maintained below 5 k*ω* while EEGs were continuously recorded at a sampling frequency of 1000 Hz and bandpass filtered from 0.5 Hz to 30 Hz. The frontal area (Fz, F3, F4, FCz, FC3, FC4), parietal region (Pz, P3, P4), central region (Cz, C3, C4, CP3, CP4), temporal region (TP7, TP8, T7, T8, P7, P8), and occipital region (POz, PO3, PO4, Oz, PO7, PO8) were the characteristic channels chosen for statistical analysis of EEGs [14]. EEGLab preprocessed the data. To eliminate industrial frequency interference, the original data were filtered in the pre-processing step with a 0.1–90 Hz bandpass filter, followed by a 50 Hz and 100 Hz trap filter [25]. Later, after rejecting defective segments and interpolating bad leads, we applied independent component analysis (ICA) to get rid of artefact components [26]. Baseline drift was ultimately removed [27] and re-referenced. After preprocessing, the artifact-free data were bandpass filtered to generate EEG signals in the *α*-band (8–13 Hz) and *β*-band (13–30 Hz); the *β*-band was then separated into three subbands: *β*1 (13–18 Hz), *β*2 (18–21 Hz), and *β*3 (21–30 Hz) for additional analysis. The power of a data segment was represented by the sum of the squares of all points in the frequency range, as given in Equation (1), where *k* denotes the number of trials in the data segment, *n* denotes the number of data points in each segment, and *x*(*k*)*_i_* denotes the value of the *i*th point in the *k*th data segment.
(1)$$E(k)=\frac{1}{n}\sum_{i=1}^{n}x(k)i^{2}$$

The same feature channels used for the EEG statistical analysis were chosen for the VEP 159 (occipital region: POz, PO3, PO4, Oz, PO7, PO8). VEP was calculated by averaging the raw 160 data for each condition, comparing them to a baseline between a 1000 ms to 0 ms before the 161 stimulus, then offline filtering the data with a band-pass filter between 0.1 and 30 hz. This 162 paper focused the study on the relevant visual ERPs components: P1, P3, and late event-163 related components, with appropriate fine-tuning of the windows for observational analysis 164 based on the results of this paper. Three time windows were selected for mean ERP ampli-165 tude analysis: Time window 1 = 80–120 ms after stimulus onset, Time window 2 = 200–320 ms after stimulus onset, Time window 3 = 800–2000 ms, and the window 3 was divided into 167 five sub-windows for analysis in 200 ms steps (window 3–1: 800–1000 ms, window 3–2: 1000–1200 ms, window 3–3: 1200–1400 ms, window 3–4: 1400–1600 ms, window 3–5: 1600–1800 ms) to 169 further observe the modulation of the late components.

## 3. Results

In this paper, the subjective data and VEP were compared by two-way repeated measures ANOVA with a within-group factor of emotion (e.g., positive, neutral, and negative) and a between-group factor of Group (desktop 2D and VR-3D). For EEG power, the role of emotions and brain regions in each visual mode were explored using two-way repeated measures ANOVA with two within-group factors of emotion (e.g., positive, neutral, and negative) and brain region (frontal, central, parietal, temporal, and occipital), and paired T-tests were conducted between 2D and VR-3D. All analyses were conducted at a 0.05 level of significance. Simple effects analysis was performed if any interaction among factors was found, and multiple comparisons were corrected with the Bonferroni method. All statistical analyses were conducted using SPSS 23.0.

### 3.1. Subjective Data

The results of subjective data are shown in Table 1. The main effect of emotional valence on the SAM scale (F(2,526) = 632.256, *p* < 0.001, *η*^2^ = 0.706) revealed that positive materials have a significantly higher valence than neutral materials (*p* < 0.001) and negative materials have a significantly lower valence than neutral materials (*p* < 0.001). The main effect of emotional arousal on the SAM scale (F(2,526) = 26.183, *p* < 0.001, *η*^2^ = 0.091) revealed that positive (*p* < 0.001) and negative (*p* < 0.001) materials evoked significantly higher emotional intensity than neutral materials. The findings demonstrated that the individuals were able to recognize the emotional connotations of the video clips, and that the emotional content might elicit greater arousal. The main effect of group on the SAM scale (F(1,263) = 93.751, *p* < 0.001, *η*^2^ = 0.263) revealed that the VR-3D mode’s valence score was significantly higher than the desktop 2D mode’s among the two visual modes (*p* < 0.001). There was a main effect of group in arousal (F(1,263) = 8.424, *p* < 0.001, *η*^2^ = 0.030) and the benefits of VR-3D remain substantial for arousal (*p* = 0.004 < 0.05).

### 3.2. EEG Data Analysis

#### 3.2.1. α. Band

The EEG power of the *α* band is shown in Figure 5. The main effect in brain regions was significant when viewing 2D (F(4,32) = 15.650, *p* < 0.001, *η*^2^ = 0.662) or VR-3D (F(4,40) = 10.365, *p* = 0.002, *η*^2^ = 0.509) emotional materials. In both visual modes, the energy was much higher in the occipital and temporal regions, particularly in the occipital region. While viewing positive (*p* = 0.012), neutral (*p* = 0.025), and negative (*p* = 0.005) materials, differences in EEG power of the occipital region elicited by different visual modes were statistically significant. Watching VR-3D videos resulted in higher EEG energy activation in the occipital region of the brain. In addition to this, VR-3D excited significantly higher *α* band waves in the frontal region (*p* = 0.030) than 2D when viewing negative material.

#### 3.2.2. *β*1. Band

The EEG power of *β*1 band is shown in Figure 6. The main effect in brain regions was significant when viewing 2D (F(4,36) = 5.130, *p* = 0.002, *η*^2^ = 0.363) or VR-3D (F(4,36) = 11.767, *p* < 0.001, *η*^2^ = 0.567) emotional materials. In the 2D visual mode, the EEG energy was significantly higher in the temporal region than the parietal region (*p* = 0.025). In the VR-3D visual mode, the EEG energy was significantly higher in the occipital region than both the parietal region (*p* = 0.023) and the central region (*p* = 0.006).

While viewing positive materials, the VR-3D mode resulted in significantly higher EEG energy in the occipital region (*p* = 0.014) compared to the 2D mode. While viewing neutral materials, the VR-3D mode compared to the 2D mode resulted in significantly higher EEG energy both in the occipital (*p* < 0.001) and frontal (*p* = 0.021) regions. While viewing negative materials, the VR-3D mode compared to the 2D mode resulted in significantly higher EEG energy in the occipital (*p* = 0.005) and temporal (*p* = 0.048) regions.

#### 3.2.3. *β*2. Band

The EEG power of the *β*2 band is shown in Figure 6. The main effect in brain regions was significant when viewing 2D (F(4,36) = 15.317, *p* < 0.001, *η*^2^ = 0.630) or VR-3D (F(4,36) = 35.547, *p* < 0.001, *η*^2^ = 0.798) emotional materials. In the 2D visual mode, the EEG energy was significantly higher in the occipital region (*p* = 0.006) and frontal region (*p* = 0.007) than the parietal region, and also the EEG energy was significantly higher in the temporal region than the parietal region (*p* < 0.001) and central region (*p* = 0.011). In the VR-3D visual mode, the EEG energy was significantly higher in the occipital region than in the frontal region (*p* = 0.031), parietal region (*p* < 0.001), and central region (*p* < 0.001). In the temporal region, the EEG energy stimulated by VR-3D was significantly higher than in the parietal region (*p* = 0.002) and central region (*p* = 0.001). Also in the frontal region, the EEG energy stimulated by VR-3D was significantly higher than in the parietal region (*p* = 0.008) and central region (*p* = 0.010). In the *β*2 band, higher EEG energy can also be stimulated in the frontal region.

While viewing positive materials, the VR-3D mode compared to the 2D mode resulted in significantly higher EEG energy both in the occipital (*p* = 0.005) and parietal (*p* = 0.039) regions. While viewing neutral materials, the VR-3D mode compared to the 2D mode resulted in significantly higher EEG energy both in the occipital (*p* = 0.005) and temporal (*p* = 0.019) regions. While viewing negative materials, the VR-3D mode compared to the 2D mode resulted in significantly higher EEG energy in the occipital (*p* < 0.001), temporal (*p* = 0.026), and frontal (*p* = 0.047) regions.

#### 3.2.4. *β*3. Band

The EEG power of the *β*3 band is shown in Figure 6. The main effect in brain regions was significant when viewing 2D (F(4,32) = 22.606, *p* < 0.001, *η*^2^ = 0.739) or VR-3D (F(4,32) = 31.130, *p* < 0.001, *η*^2^ = 0.796) emotional materials. In both visual modes, the energy was much higher in the occipital, temporal, and frontal regions.

While viewing positive materials, the VR-3D mode compared to the 2D mode resulted in significantly higher EEG energy both in the occipital (*p* = 0.005) and temporal (*p* = 0.045) regions. While viewing neutral materials, the VR-3D mode compared to the 2D mode resulted in significantly higher EEG energy in the occipital (*p* < 0.001), temporal (*p* = 0.037), parietal (*p* = 0.049), and frontal (*p* = 0.007) regions. While viewing negative materials, the VR-3D mode compared to the 2D mode resulted in significantly higher EEG energy in the occipital (*p* = 0.005), parietal (*p* = 0.037), and frontal (*p* = 0.048) regions.

### 3.3. Brain Topography

Warm tones denote high brainwave activity and cold tones denote low brainwave activity in Figure 7, which is a brain topographic map based on the average energy of all individuals. According to the energy distribution of the brain topographic map, visual stimulation increased activity in the frontal, occipital, and temporal regions by VR-3D. It is worth noting that frontal areas were more active in all brain regions in *β*2 and *β*3 frequency bands, especially in the VR-3D visual mode.

### 3.4. VEP Data Analysis

For each of the three selected time windows (including sub-windows for Window 3), a two-way repeated measures ANOVA was conducted on emotion and visual modes group factor for each time window datum as shown in Figure 8.

In the first time window (80–120 ms), the results showed both main effects of emotion (F(2,18) = 6.659, *p* = 0.007, *η*^2^ = 0.425) and group (F(1,9) = 24.477, *p* = 0.001, *η*^2^ = 0.731), which revealed that negative stimuli induced a significantly larger positive component than neutral stimuli (*p* = 0.004) and VR-3D induced a significantly larger positive component than 2D (*p* = 0.001).

In the second time window (325–425 ms), the results showed a main effect of emotion (F(2,18) = 8.084, *p* = 0.003, *η*^2^ = 0.473), which revealed that positive stimuli induced a significantly larger positive component than negative stimuli (*p* = 0.039). However, visual modes group factor was not found to make a significant difference in the second window.

The third time window (800–1800 ms) is divided into five sub-windows to be analyzed individually. In window 3-1 (800–1000 ms), the results showed a main effect of emotion (F(2,18) = 3.665, *p* = 0.056, *η*^2^ = 0.289), which revealed that positive stimuli induced a significantly larger positive component than negative stimuli (*p* = 0.029) and a lack of significant difference induced by visual modes group factor. Neither group nor emotion factors were found to have an effect in window 3–2 (1000–1200 ms). In window 3–3 (1200–1400 ms), the main effect of group (F(1,9) = 8.158, *p* = 0.019, *η*^2^ = 0.475) showed that energy values were significantly higher for VR-3D than 2D (*p* = 0.019); the mean difference is 3.812 (95%CI: 0.793–6.832). In window 3–4 (1400–1600 ms), the main effect of group (F(1,9) =8.710, *p* = 0.016, *η*^2^ = 0.492) showed that energy values were significantly higher for VR-3D than 2D (*p* = 0.026); the mean difference is 4.004 (95%CI: 0.935–7.073). In window 3–5 (1600–1800 ms), the main effect of group (F(1,9) = 10.314, *p* = 0.011, *η*^2^ = 0.534) showed that energy values were significantly higher for VR-3D than 2D (*p* = 0.011); the mean difference is 4.406 (95%CI: 1.302–7.509). In addition, it is interesting to note that the significant difference induced by visual modes becomes stable and tends to increase after 1200 ms.

### 3.5. VEP Sources Analysis

The findings of the VEP analysis showed a main effect of group within the 1200–1800 ms window, and the difference caused by visual modality was substantial. We used the sLORETA traceability method in the LORETA software to compare the current density distribution and the density intensity of brain activation areas under stimulation in the two control groups (2D and VR-3D). This served as the foundation for the examination of the window’s positive slow waves.

The middle frontal gyrus is where the current density of VEP sources differs most compared to 2D and VR-3D, as seen in Table 2 and Figure 9. The findings showed that stereoscopic visual stimuli in the VR environment are mostly responsive in higher cognitive regions, and that the significant wave amplitude increase in the 1200–1800 ms window may represent a further processing of cognitive, attentional, and visual information, thereby responding to the visual cortex of the human brain. 

## 4. Discussion

### 4.1. Subjective Rating

Subjective data is a conscious self-report, and part of the brain’s processing of emotional stimuli can differ from the participant’s conscious evaluation [27]. In this experiment, the results of the SAM subjective scale showed that the subjects were able to understand the emotional information conveyed by the video in a short period of time and accurately identify the emotional valence of the video, whether it was a 2D desktop video or a VR-3D video. In particular, emotional materials were more evocative than neutral materials under subjective perception. However, in the VR-3D environment, subjective scales indicated that subjects felt more obvious differences in emotional valence and more intense emotional arousal.

In summary, the results of subjective rating indicated that VR-3D is significantly more capable of communicating different emotions and emotional arousal in terms of subjective perception.

### 4.2. EEG Results

EEG activity has different features in different brain regions and represents different functions. Simple visual feature detection is possible in the occipital visual cortex, while complex feature motion discrimination is mostly performed in the temporal lobe [28,29]. The greater levels of sensory processing, linguistic abilities, and spatial awareness are all attributed to the parietal areas [30]. According to research, the frontal area is the focus of higher cognitive functions. Complex neurological processes such as perception, thought, attention, feedback, and other higher activities are typically found in the region where the prefrontal cortex meets the occipital, parietal, and frontal cortices [31,32,33,34]. The *α* waves of EEG signals may reflect visual fatigue to some extent [35], while the *β* waves are often used to reflect emotional arousal states and cognitive processes [36].

The experimental results showed that, in terms of the number of brain areas where there were significant differences caused by visual modes, the differences are more widespread in the high-frequency band. The high-frequency domain was the main waveform that occurred when the cerebral cortex was excited and the central nervous system was highly active. It is concluded that, in the high-frequency domain, the higher energy brought by VR-3D in more brain regions proved that VR-3D produced stronger emotional arousal and empathy effects in the human brain. Of all the frequency bands analyzed, the occipital region, as the cortical layer associated with vision, was unexpectedly sensitive to stereoscopic vision regardless of emotion. Yet, the frontal region was more likely to show significant differences caused by stereo vision when viewing negative stimulus materials. It can be hypothesized that the negative material in the VR-3D environment brought about stronger emotional arousal, which led the subjects to concentrate more during the intense stimulation.

From this it can be concluded that stereoscopic vision in virtual reality environments probably can influence perception to a greater extent, leading to greater emotional arousal, where negative emotions have a greater impact on higher cognitive areas of the brain.

### 4.3. VEP Results

The visual processing part of the brain, the occipital region, also allows for the monitoring of emotional modulation processes [22,37]. In the early P1 component window 1, the emotional material inspired a greater positive component, particularly significant for the negative stimulus materials compared to the neutral materials. In the next time windows, until window 3-2, the positive stimulus material induced a greater wave amplitude, yet in the subsequent windows (after 1200 ms) the difference in emotion brought about was not significant. This may reflect the fact that emotional content increases activation of the external visual cortex, leading to modulation of the P1 component. Negative emotions were more intense and more likely to focus the attention of the subjects during the primary phase of early visual processing, whereas the difference may not be significant in the middle and late phases because of brain emotion regulation mechanisms. The results from analysis of the difference caused by stereo vision in window 1 showed that a significantly higher potential was produced in VR-3D. However, it was not until after window 3-3 (after 1200 ms) that the greater amplitude triggered by the VR-3D environment compared to 2D became significant and tended to increase gradually. From this, it was presumed that the immersion brought about by stereoscopic vision can influence early selective visual processing. Nevertheless, the significant differences in event-related components that only became apparent late presumably reflected the course of identifying with the presence of the surroundings and creating a sense of immersion during viewing. Higher cognition from stereoscopic vision may require at least 1200 ms to become consistently stable, and it can be presumed that the spatio-temporal homogeneity of VR with the real world is identified after this point in time.

It can be concluded from the VEP analysis that both stereo vision and emotion are factors that may influence early visual processing at the moment of stimulus onset. A consistent perception of the effects of VR-3D on emotional arousal and attention allocation occurred at 1200 ms after the stimulus began, probably related to the time it takes for a person to identify with the spatio-temporal homogeneity of the VR environment with the real world.

### 4.4. VEP Sources Results

It has been shown that the prefrontal cortex is associated with attention-dependent modulation of neural activity in the visual association cortex (VAC) [38] and that its importance in the spatial localization of visual stimuli is well established [39]. In addition to this, the functional interaction between visual areas and frontal cortex may contribute to conscious vision [40], and VEP sources show that the differences brought about by stereoscopic vision occur mainly in frontal areas, rather than in the occipital region where visual processing takes place. It can be hypothesized that the effect of stereo vision on emotional arousal relies on a network of multiple functional fields in the occipital and prefrontal cortices.

The results of the VEP sources showed that higher cognitive areas respond to VR visual stimuli primarily after 1200 ms, which may signal the start of the regulation of attention allocation to conscious perception of objects and further processing of stereo visual information, thereby responding to the visual cortex areas of the human brain.

## 5. Conclusions

Due to their own visual style, VR films offer a different sense of immersion and reality than traditional movies. In order to better understand the rhythm of how to guide the audience’s emotional changes, it is necessary to approach them through a more scientific method. The emergence of neurocinematography links VR cinema to cognitive neuroscience, opening the way to explore the science of VR cinema based on access to physiological data. In this paper, objective EEG data and subjective data were collected based on neurophysiology with the exclusion of possible different gain effects from audio, and differences in emotional arousal processes between desktop 2D and VR-3D were compared by analyzing EEG energy features, VEP, and subjective ratings.

The results demonstrated that stereoscopic vision in a virtual reality environment can trigger increased emotional arousal. This conclusion was supported by both subjective perception and objective physiological signals. EEG analysis of differences between brain regions yielded the conclusion that negative emotions have a greater effect on higher cognitive areas of the brain. VEP analysis further expanded the temporal dimension and found that both stereoscopic vision and emotion are factors that may affect early visual processing. The VEP sources analysis also hypothesized that the collaboration between frontal and occipital lobes after about 1200 ms may indicate the recognition of the spatiotemporal homogeneity of the VR environment with the real world, the development of conscious perception of stereoscopic space, and further processing of visual information.

## Figures and Tables

**Figure 1 sensors-22-08491-f001:**
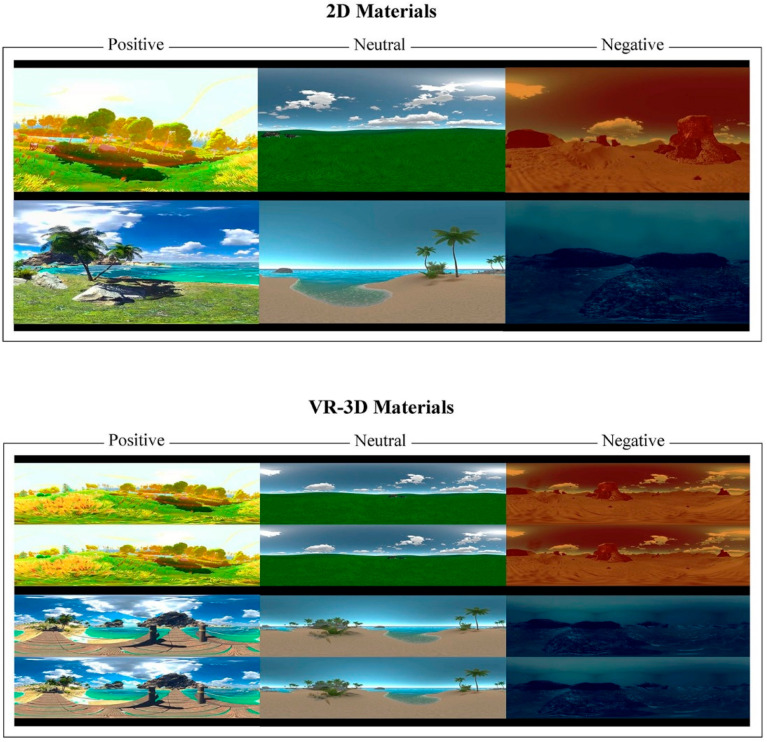
The experimental materials were separated into two groups: 2D and VR-3D. There were 24 various environments in the positive, neutral, and negative VR video materials.

**Figure 2 sensors-22-08491-f002:**
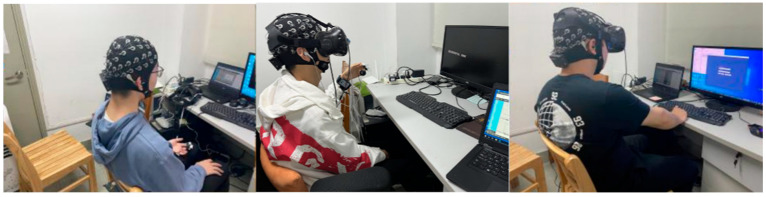
Experimental environment and equipment connections.

**Figure 3 sensors-22-08491-f003:**
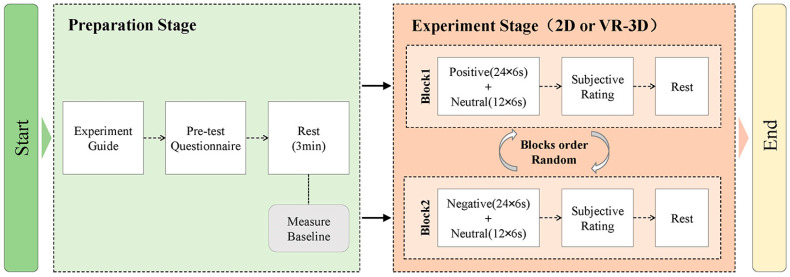
Flowchart for the cognitive psychology experiment.

**Figure 4 sensors-22-08491-f004:**
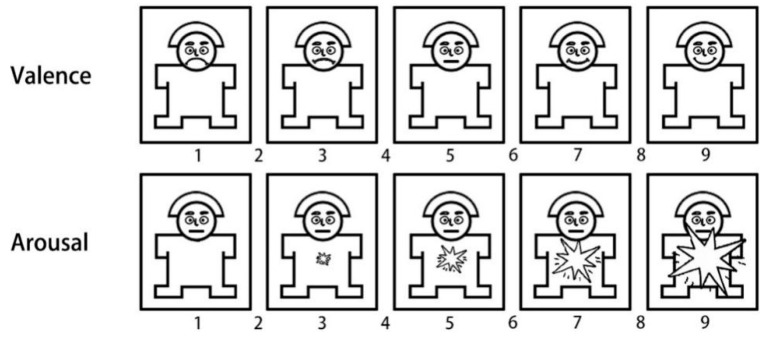
Schematic diagram of the SAM (Self-Assessment Manikin) scale.

**Figure 5 sensors-22-08491-f005:**
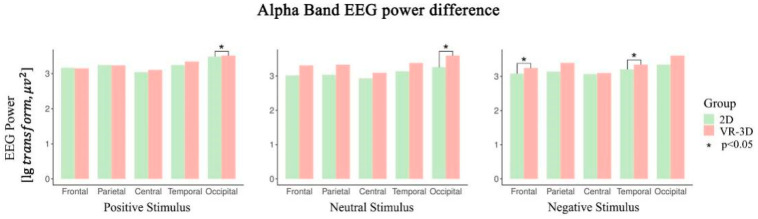
The EEG power of *α* band.

**Figure 6 sensors-22-08491-f006:**
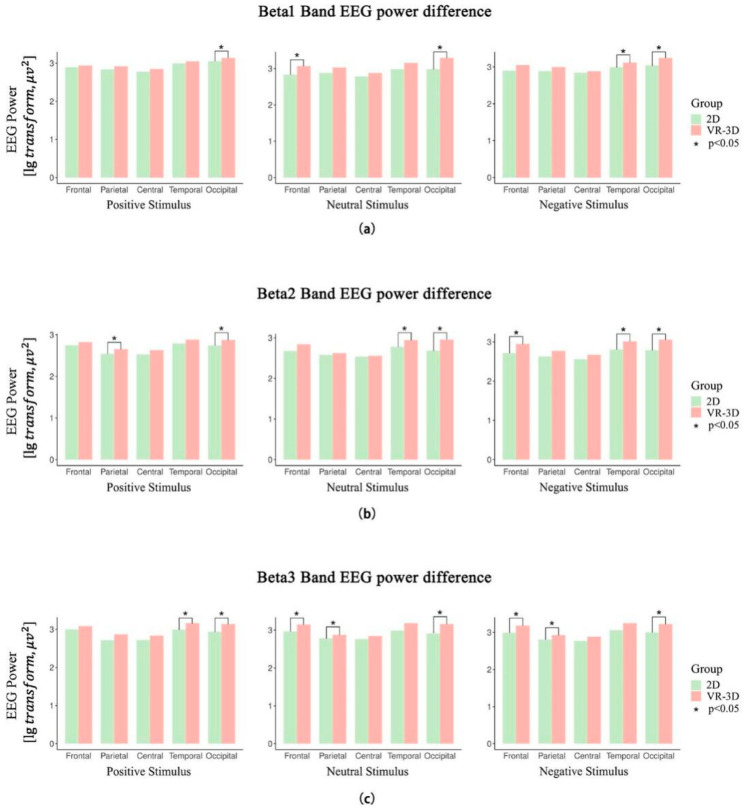
The EEG power of *β* band. (**a**) The distribution of EEG waves in brain regions with the *β*1 band. (**b**) The distribution of EEG waves in brain regions with the *β*2 band. (**c**) The distribution of EEG waves in brain regions with the *β*3 band.

**Figure 7 sensors-22-08491-f007:**
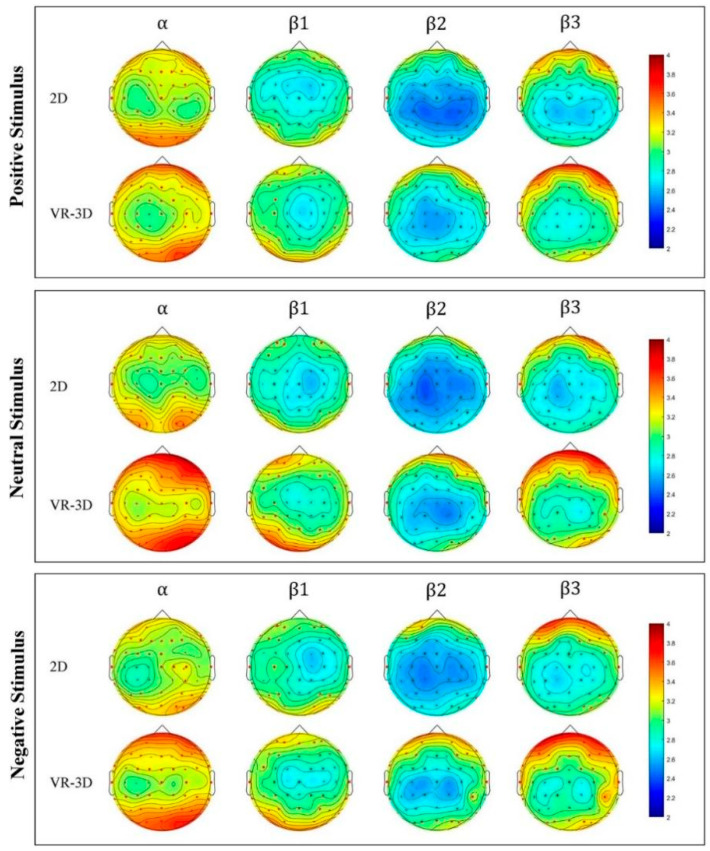
Brain topography evoked by different videos.

**Figure 8 sensors-22-08491-f008:**
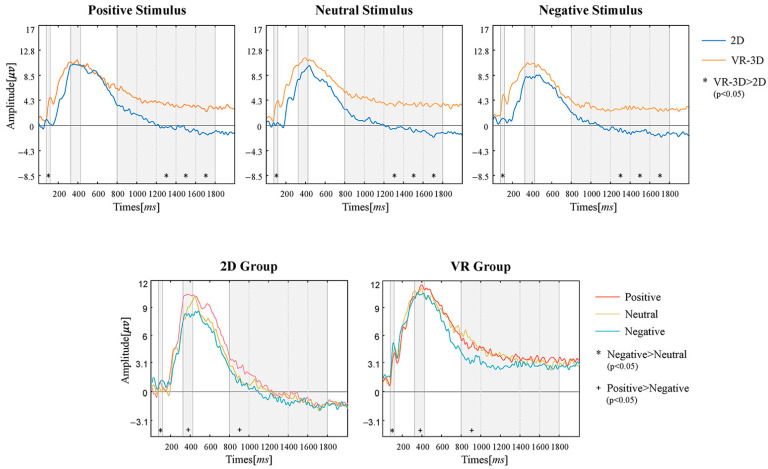
VEP Result.

**Figure 9 sensors-22-08491-f009:**
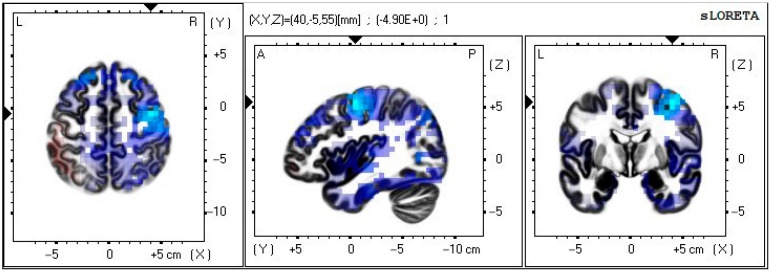
Schematic of current source density distribution for visual modes differences.

**Table 1 sensors-22-08491-t001:** The average value (mean ± S.D.) of valence and arousal in SAM (16 volunteers).

Group	Score Type	Positive	Neutral	Negative
2D VR-3D	Arousal	6.00 ± 1.08	5.06 ± 0.84	3.17 ± 1.36
	Valence	4.89 ± 2.14	4.20 ± 2.17	5.01 ± 2.21
	Arousal	6.39 ± 1.25	5.59 ± 1.43	3.87 ± 1.07
	Valence	5.03 ± 1.87	4.63 ± 1.85	5.21 ± 2.01

**Table 2 sensors-22-08491-t002:** Statistical comparison of VEP source current density between 2D and VR-3D.

Talairach Coordinate (TAL)			Brodmann Area	Lobe	Structure
**X**	**Y**	**Z**	6	Frontal Lobe	Middle Frontal Gyrus
40	−2	51			

## Data Availability

The data presented in this study are available on request from the corresponding author. The data are not publicly available because we are creating an EEG data set.

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
