# Peer review of "A Comparative Study on the Temporal Effects of 2D and VR Emotional Arousal"

_sensors, 2022, doi:10.3390/s22218491_

Round 1

Reviewer 1 Report

Better describe ratings using stereoscopic vision in virtual environments,

Explain why the effects of VR-3D on emotional arousal and attention  allocation occurred at 1200 ms after the stimulus begins. What about comparison with other literature data?

Does EEG energy features are only measure technique?

What about others measurement?

Visual evoked potentials are not explored properly.

Author Response

We are very grateful for your suggestions. Please see the attachment for the corresponding responses and amendments.

Reviewer 2 Report

1. It is suggested to present your research questions in from of a research motivation statement. E.g. #1 Stereoscopic visual stimuli is common in VR, however the research about its effect on visual processing is still lacking. In addition, correlation between stereoscopic visual stimuli with the changes in emotion is an important characteristic to be explored. --sorry if i have misinterpreted your RQ.

2. Material and methods section is well presented.

3. On page 5, section 2.4. – I think the first sentence about Bradley and Lang created SAM in unnecessary. You may tell readers straightaway about the method you have chosen, if Figure 4 belongs to them (Bradley & Lang), you may need permission from them.

4. I strongly recommend combining the result and discussion as one section. Both sections are well presented and written, thus putting them together may ease the readers in following the story.

5. If possible, provide a brief discussion regarding the relation between the results – Subjective Rating-EEG-VEP. You may put this at the end of result and discussion section, or in the conclusion.  

Author Response

(The authors gave the same response as above.)

Round 2

Reviewer 1 Report

Authors responded to my comments.

Author Response

Thank you very much for your reply and your confirmation. We have checked the English spelling and adjusted some of the paragraph expressions in the light of your comments. You can see the specific changes in the attached document.
